# The Motion Paradigm of Pre-Dock Zearalenone Hydrolase Predictions with Molecular Dynamics and the Docking Phase with Umbrella Sampling

**DOI:** 10.3390/molecules28114545

**Published:** 2023-06-04

**Authors:** Xi-Zhi Hong, Zheng-Gang Han, Jiang-Ke Yang, Yi-Han Liu

**Affiliations:** 1Pilot Base of Food Microbial Resources Utilization of Hubei Province, College of Life Science and Technology, Wuhan Polytechnic University, Wuhan 430024, China; donmega12357@gmail.com (X.-Z.H.); zhengganghan@whpu.edu.cn (Z.-G.H.); 2Key Laboratory of Industrial Fermentation Microbiology, Ministry of Education, The College of Biotechnology, Tianjin University of Science and Technology, Tianjin 300453, China

**Keywords:** zearalenone hydrolase, neural relational inference, umbrella sampling, MMPBSA, bmDCA

## Abstract

Zearalenone (ZEN) is one of the most prevalent estrogenic mycotoxins, is produced mainly by the Fusarium family of fungi, and poses a risk to the health of animals. Zearalenone hydrolase (ZHD) is an important enzyme capable of degrading ZEN into a non-toxic compound. Although previous research has investigated the catalytic mechanism of ZHD, information on its dynamic interaction with ZEN remains unknown. This study aimed to develop a pipeline for identifying the allosteric pathway of ZHD. Using an identity analysis, we identified hub genes whose sequences can generalize a set of sequences in a protein family. We then utilized a neural relational inference (NRI) model to identify the allosteric pathway of the protein throughout the entire molecular dynamics simulation. The production run lasted 1 microsecond, and we analyzed residues 139–222 for the allosteric pathway using the NRI model. We found that the cap domain of the protein opened up during catalysis, resembling a hemostatic tape. We used umbrella sampling to simulate the dynamic docking phase of the ligand–protein complex and found that the protein took on a square sandwich shape. Our energy analysis, using both molecular mechanics/Poisson–Boltzmann (Generalized-Born) surface area (MMPBSA) and Potential Mean Force (PMF) analysis, showed discrepancies, with scores of −8.45 kcal/mol and −1.95 kcal/mol, respectively. MMPBSA, however, obtained a similar score to that of a previous report.

## 1. Introduction

Zearalenone (ZEN), produced by mold fungi mainly belonging to the *Fusarium* family, is a prevalent grain toxin in agriculture today. ZEN is a xenoestrogen that binds to estrogen receptors of cells, leading to a hormonal imbalance in the body, which may in consequence lead to numerous diseases such as prostate or ovarian cancers. ZEN pollution generates significant economic losses and poses a substantial health risk to humans and cattle [1,2,3]. Research on ZEN removal procedures is an important topic that has garnered serious attention. Currently, there are three types of methods for degrading ZEN: physical degradation, chemical degradation, and biodegradation. Common physical degradation methods include activated carbon adsorption or direct pressure cooking from 120 °C to 140 °C to break down the ZEN structure [4]. Common chemical methods include super-oxidation of ozone [5], peroxidation of hydrogen peroxide [6], and oxidation of didodecyldimethylammonium bromide [7]. Biological degradation of ZEN supplies green and environmentally friendly measures and alternatives to high-energy-consuming methods and chemical reagents. As a result, an increasing amount of effort is focused on developing strategies to employ microbes and their enzymes. Among all the enzymes known to be capable of degrading ZEN, zearalenone hydrolase (ZHD) has received great attention for its specificity and high efficiency. Since the discovery of ZHD101 [8], which can achieve complete degradation in a relatively brief duration, and Is by far the most potent ZEN-degrading enzyme found in nature, other types of hydrolases such as those found in *Trichoderma aggressivum* [9], *Rhinocladiella mackenziei* [10], *Pseudomonas putida* [11], and *Fonsecaea monophora* [12] have been discovered.

To improve the degrading efficiency of hydrolase on zearalenone and enhance its stability to the reaction temperature, methods of rational design have generally been used. Within rational design, molecular mechanics (MM) descriptions of enzymes provide fundamental knowledge of the molecular structure and conformational changes when reacting with substrate molecules and environmental factors. By constructing a protein dynamics simulation of hydrolase ZHD607 from *Phialophora americana*, the mutant experiments on the plasticity of loop 136–142, which is structurally located opposite the entrance to the pocket of ZHD607, the enzymatic activity was improved [13].

Due to the lack of detailed knowledge about the structural changes in the inter-substrate binding of zearalenone hydrolase and the specific binding mode of the active center of the substrate, rational design and molecular modification of these ZHD molecules are lagging behind. The motion mechanism, conformational change, and movement locus of enzymes may interfere with the activity and thermal stability. To study the actual molecular state of the ZHD, we will perform a molecular dynamic simulation of the representative enzyme, explore the potential allosteric sites, and reveal the correlation in multi-state ZHD structure along with some stringent residues that could deliver extensive effects on the molecular motion mechanism. We will also use umbrella sampling to discover the pathways during catalytic reaction to identify the critical residues when the enzyme contacts the substrate.

## 2. Results

### 2.1. An Evolutional Analysis of Hydrolase and the Hub-Gene Definition

A total of 452 hydrolase sequences were obtained from two iterations of the Position-Specific Iterated BLAST (PSI-BLAST) algorithm and used as the initial dataset. A pipeline was used to narrow the candidates down to 384 sequences with a length ranging from 240 to 300 amino acids. These 384 sequences underwent iterative BLASTp analysis, and the identity for each sequence was calculated using a gradient of 60%, 70%, 80%, 90%, and 95%. A violin plot was used to identify hub sequences based on their identity (Appendix A). As shown in Appendix A, the number of representative sequences gradually decreased as identity increased from 60% to 95%. When the identity was above 60 degrees and at 80%, eight representatives remained.

To gain a better understanding of the differences between the hub genes, Boltzmann machine learning direct coupling analysis (bmDCA) was used to analyze the correlation between various residues, bmDCA analysis was performed for eight hub genes, and the consensus sequence among them was found to be dcaZHD. Multiple sequence alignment revealed the local functional sequences: LIP in β3, GMSRS in the loop between the cap domain and the hydrolase domain, and SSGA in α3. RmZHD from *R. mackenziei*, NbZHD (GenBank ID: WP_138717259) from *Nonomuraea basaltis*, and SbZHD (GenBank ID: WP_111491235) from *Streptacidiphilus bronchialis*, which were selected for the following MD simulations, were aligned with dcaZHD to inspect the conservative sequence (Figure 1A). RmZHD was also aligned with the eight hub genes. LIP, GMSRS, and SSGA were also identified as conservative sequences (Figure 1B).

### 2.2. Extensive Simulation of Molecular States

The 1 μs simulation time scale encompasses both binding and releasing of the enzyme to the substrate, which breaks the energy barrier before separating from ZHD. RMSD analysis was employed to assess the motion range of ZHD, revealing the essential stability of NbZHD (Figure 2A), indicated by the higher, flatter RMSD line than that of SbZHD, which suggests less stability. The RMSF analysis shows that the enzyme-activity-related atom range distinguishing SbZHD from NbZHD is from around 2000 to 3300 atoms, beyond which the better correspondence implies that the molecular extensive motion of residues 139 to 222 contributes to degrading ZEN (Figure 2B). Following the 945 ns simulation, the ligand had concatenated into the cavity of ZHD, which marks the initial stage of the subsequent pulling process.

To further investigate the motion mechanism, a deep learning-based graph network was used to discover the pathway of ZHD. The orange spheres represented each residue in the pathway, which may contribute to the catalytic and motion mechanisms. The upper grey domain pertains to the cap domain, while the teal domain belongs to the α/β hydrolase domain. As seen in Figure 3A,B, a pathway consisting of eight residues across the cap domain provides a novel insight into this enzyme’s motion mechanism. Another pathway also contains seven residues, covering only the forehead of the cap domain, but one residue at the end differs from the previous. This pathway discloses a possible low-fluctuation motion mechanism, whose probability can reach up to 0.9999. In this case, the seven residue pathways may reveal the mechanism of ZHD in motion. Table 1 illustrates several possible pathways consisting of seven residues, with possibilities exceeding 99.9%, and two eight-residue pathways. Based on pathways 1 and 2, the eight residues pathway always begins with Y209, R139, V145, M155, P174, A176, and L178 but differs at the end, which is either A186 or Q183. However, the varying residues at the end will not alter the essentiality of the motion mechanism. Both eight-residue pathways completely open up the cap domain to the oblique top in the opposite direction. The seven-residue pathway starts from the midpoint of the cap domain (Figure 3C), revealing a more realistic entrance to ZEN that is tighter than the prior pathways in both sense and probability, beginning with Q183, E177, P174, L154, L149, and Q146 and ending with I202.

### 2.3. Motion Paradigm from Umbrella Sampling

In pursuit of an accurate protein motion paradigm, we performed umbrella sampling of 98 windows to investigate the pulling process, commencing at the reaction coordinate of −0.6 nm for 8 Å. Before the production run, protein conformation was constrained, and each window was equilibrated for 100 ps by NPT. Each window was then simulated for 5 ns, and the whole simulation lasted 490 ns. The curve met the local minima at a reaction coordinate of −1.257 nm, which marked the docking phase. Below the reaction coordinate of −0.648 nm, the PMF decreased gradually, indicating the ligand’s entry into the cavity (Figure 4). The structure at the inflection point defined the boundary between docking and receiving the ligand, signifying that the binding energy growth rate had reached the maximum. The lowest energy conformation emanated from the structure to the left down, representing the docking phase at −1.25 nm (State 1). The structure at −0.75 nm (State 3) showed the ligand at the protein’s gate, with the PMF being a local maximum. The structure situated at the center of the PMF analysis was at −1.02 nm (State 2) and had energy of −2.23 kcal/mol. The PMF showed that the binding free energy of the docking process was −1.95 kcal/mol, with the PMF graph starting from −2.6 kcal/mol at the catalytic center and progressing to −0.65 kcal/mol at the gate.

Based on the essential motion of key residues in the long-range period, we presumed that ZEN moves linearly into the catalytic center and follows the same path out. An NRI search for the most compact region between the center-of-mass (COM) of the ligand and protein was conducted in the equilibrated state. The 6th and 97th windows were concatenated to form one consistent window, and the production run of these two windows was rerun from the NPT equilibrated state for another 200 ns. As the ligand had adapted to the catalytic core, the system was situated at the potential well, making the ligand work like a Trojan to be released. A total of 433 frames from the 7th window and 926 frames from the 97th window constituted the training data for the NRI model. The NRI model highlighted residues for ligand release and incorporation in this shorter duration. The motion residues of the cap domain started from M207, R139, and V145 and ended with Q152, while the α/β hydrolase started from D45, A38, and S62 and ended with L66. The backside of ZHD showed four motion residues from each domain, and the front side of ZHD contained the ligand (Figure 5).

### 2.4. Alanine Scanning Mutations for Inspecting Energy Shifting

To validate our assumptions, we executed alanine mutations to minimize any potential ionic connection creation in the pathway and determine whether the motion mechanism is natural. We selected R139 from the cap domain and D45 from the hydrolase domain, as they are motion residues of each domain that contain one charged amino acid. We used MMPBSA (molecular mechanics with Poisson–Boltzmann and surface area solvation) for alanine mutation, which is based on the binding energy calculation from molecular dynamics sampling. As the charge of the residue decreases, both mutants exhibited the diverse characteristics of ΔG_complex_. The substantially increased ΔG_complex_ of D45A might damage its enzymatic stability, while R139A showed the opposite (Figure 6A,B). The ΔG_bind_ of D45A is consistent with the wild type, while the ΔG_bind_ of R139A is more stable (Figure 6C,D). Through alanine scanning, we obtained the binding free energy of NbZHD twice, as listed in Table 2.

## 3. Materials and Methods

### 3.1. Lactone Hydrolases Co-Evolutionary Relation Analysis

We selected the zearalenone hydrolase ZHD607 as a seed sequence to be blasted against the NCBI database using the Blastp program (https://blast.ncbi.nlm.nih.gov/Blast.cgi accessed on 13 May 2023). We then performed multiple sequence alignment using ClustalW and ClustalOmega software (http://www.ebi.ac.uk/clustalW accessed on 13 May 2023) to identify the hub genes. This alignment was later delineated using ESPript software (https://espript.ibcp.fr/ESPript/cgi-bin/ESPript.cgi accessed on 13 May 2023) [14].

### 3.2. Molecular Dynamics Analysis of Lacton Hydrolase

We predicted the structure of ZEN hydrolases using Uni-fold of Hermite. Implementing the charmm36-jul2021 force field for our molecular dynamic simulation, we built the topology of the ligand ZEN with CGenFF [15] and later coupled it to our ZHD system as a compound. We used the steepest descent minimization to reduce the system, and we applied a particle number, volume, and temperature equilibrium (NVT) system to stabilize the temperature system for 1 ns, with its coupling method being V-rescale. We used the particle number, pressure, and temperature equilibrium (NPT) system to minimize the pressure coupling error for 1 ns, and its temperature coupling method was V-rescale, and the pressure coupling method was Berendsen. The system’s temperature coupling was attached at 310 K, the optimal temperature for most enzymes of this category. We performed MD for 1 μs, and the temperature and pressure coupling method were similar to an NPT run. We analyzed the trajectory by using gmx trjconv, while RMSD and RMSF analyses were performed using gmx rms and gmx rmsf software [16].

### 3.3. Umbrella Sampling

For Umbrella Sampling, we selected one specific frame where ZEN was near the cavity of ZHD for the pulling process. The pulling force was applied along the direction (−0.5, −0.5, 0.5), forcing ZEN to go deeper into the cavity for 500 ps. The system was coupled using V-rescale at a temperature of 310 K, and the pressure coupling was Parrinello-Rahman. Only h-bonds of ZHD were constrained. The pulling rate of change of the reference position was 0.01 nm/ps, and we set the harmonic force constant at 1000 kJ·mol^−1^·nm^−2^. Trajectory analysis selected 1250 frames for distance analysis (by gmx distance), sieving 98 windows for the succeeding simulation. After the pulling process, we performed 100 ns of NPT simulation to obtain an equilibrated state. During the NPT simulations, the temperature coupling was V-rescale, and the pressure coupling was C-rescale. The production run of each window was 5 ns. Potential Mean Force (PMF) analysis was performed using the Weighted Histogram Analysis Method (WHAM) [17], and we selected the 6th and 97th windows out of 98 for further analysis for 200 ns.

### 3.4. Structural Pathway and Energy Analysis

We used neural relational inference (NRI) (https://github.com/juexinwang/NRI-MD accessed on 13 May 2023) to mine the underlying pathway of hydrolase and learn its long-range allosteric interactions [18]. For alanine scanning, we used gmx MMPBSA (https://github.com/Valdes-Tresanco-MS/gmx_MMPBSA accessed on 13 May 2023) [19,20], using the 5th PBRadii and a temperature of 310 K. PB method was used along with rediopt as 0, istrng as 0.15, and fillratio as 4. Second structure analysis was conducted by gmx do dssp and gmx density.

## 4. Discussion

The deconstruction of ZEN by hydrolases is attributed to a few enzyme candidates, among which ZHD is the most efficient [8]. A novel enzyme, RmZHD, which is more thermally stable than ZHD101, was discovered from the organism *R. mackenziei* CBS 650.93. Electrostatic influence analysis shows that three residues, Asp34, His128, and Val130, located at the catalytic core of ZHD, make significant contributions to the synergistic reaction. While Asp34 and His128 can increase the energy barrier and impede the reaction, Val130 plays the opposite role [21]. Apart from residues that interact with ZEN, the allosteric residues in the protein can affect the deconstruction of ZEN. Identifying the allosteric pathway provides insight into the mobility pattern of the protein. In light of this, we developed a program with the help of NRI to identify the allosteric pathway of the specific protein family targeting particular substrates. We selected eight hub genes that represent the family’s general features and then used extensive 1 μs simulation and umbrella sampling to determine the allosteric pathway of separate stages. Finally, alanine scanning analyzed the energy shift in charged polar residues in the allosteric pathway to reveal the residues critical to altering energy barriers. The findings of these analyses decipher the deconstruction of ZEN by hydrolases and bear potential implications for future research in this field.

### 4.1. Hub Genes and MSA Survey the Extensive Character of Protein

A higher sequence identity indicates a greater sequence similarity. Sequences that have a high number of instances above a specific identity level are more representative, and hub genes can be derived from them. The degree of a given sequence is quantified as the number of sequences present above the specified identity level [22]. However, selecting the appropriate identity level can be challenging. Using a 60% or 70% identity threshold may result in hub genes that represent more sequences from the family but with an excessive number of representative sequences. Conversely, a 90% or 95% identity threshold may result in hub genes that represent fewer sequences from the family. Therefore, eight hub genes from the 80% identity threshold were chosen for further study, as they represent a comparatively higher number of sequences. Among 384 sequences, however, the predicted residue pathways are comparatively evolutionarily conserved as they merely provide one possible perspective on the dynamic paradigm of the ZHD; minor local changes could result in different residue pathways (Appendix A). Specifically, the pathways derived from the docking phase represent a potential logic for the movement during the actual catalytic process, whereas those from before the docking phase suggest a rationale for how the ZHD recruits the ZEN.

### 4.2. Umbrella Sampling and Molecular Dynamics Inspect Conformation Veer

By utilizing umbrella sampling and molecular dynamics simulation, we employed the NRI model to identify allosteric residues across various stages. The NRI model utilizes GNN [23] to learn the network dynamics’ embedding by constraining the training error between the reproduced and simulated trajectories. The derived embedding separates the crucial functions of the residues throughout the conformational shift, providing insights into the mechanisms of protein allostery. The realistic motion paradigm of NbZHD can be depicted based on the allosteric pathway. Initially, the ZEN flows around the ZHD while the cap domain of the protein unwinds like a hemostatic tape along the motion residues pathway. Once NbZHD captures the ligand, the catalytic reaction commences. From a general perspective, NbZHD represents a box with a dual compartment (Figure 7). The docking of the ligand is followed by the counterclockwise rotation of the cap domain and the clockwise rotation of the hydrolase domain, as depicted in the docking statement. After the reaction, NbZHD operates like a square sandwich, whereby the upper compartment rotates to the left, and the lower compartment rotates to the right. The former paraphrasing refers to a motion pathway of structural movement during protein catalysis, which probably also involves the function of the protein. Although the concatenated simulation exhibits a plunge curve of RMSD between the receiving and docking states, higher vibrancy is noticeable before the ZEN docks (Appendix A). Nonetheless, the role of ZEN has been overlooked so far; thus, what is its trajectory? We used the morph program by Pymol for analysis (Appendix A). Appendix A illustrates the process of the ZEN beginning to rotate in the course of departure, while Appendix A showcase the ZEN traversing through the side view and front view, respectively. Contradicting our initial assumption, the ZEN does not get ejected from the cavity of NbZHD but emerges through the aperture of the hydrolase domain. From the latter two videos, we observed that the aperture is located in the middle of the backside of the α-helix, i.e., α3 and α4. The ZEN passes through the tunnel between the loop 30–37 and the head of α4 (residues 102 and 103) before crossing the middle of α3 (residues 75–89) and α4 (residues 102–113) [24].

### 4.3. Sub-Protein and Energy Analysis Conveys Detailed Conformation Shifting

Analyzing the structure of D45 and R139, in conjunction with two mutants, A45 and A139, could provide insights into why neither D45 nor R139 should be mutated. The carboxyl group of the D45 side chain can create an electrostatic force that attracts the M1 backbone amino, creating a distance of 2.6 Å (Figure 8A,B). However, the replacement of D45 with alanine (D45A) cannot support a similar or shorter distance of the electrostatic force that can contribute to pulling the hydrolase domain along the pathway D45 → A38 → S62 → L66. This applies to R139 as well. On the other side of R139, the electro-negativity of the sulfur atom in M207 can attract the amino of the R139 sidechain at a distance of 3.8 Å (Figure 8C,D). However, the substitution of arginine with alanine (R139A) can only offer the shortest electrostatic attraction distance of 5.7 Å. The second structural changes were examined using the dssp program of GROMACS. A comparison was made using Figure 9(Aa), which is a plot of the last 500 ns of a long-range simulation, and Figure 9(Ab), which is a plot of a concatenated simulation from two different windows. During the long-range simulation, the ligand often collapses in the potential well. Examining the comparison between the two simulations helps identify the residues that conform to different structures during the catalytic simulation. Residues D170 to R190 exhibit different structures in the two simulations. While these residues are distributed evenly in the long-range simulation, regardless of mass or charge, they huddle together in the concatenated simulation in the late 4 nm (Figure 9B,C). These residues can be regarded as a channel that provides a tunnel through which the ligand passes to the hydrolase domain. Therefore, residues D170-R190 are named as the bolt in the following discussion. Based on the data presented in Appendix A, the bolt in the unbound state (colored violet-purple) is less tight than that in the State 2 (colored slate), leaving a narrower channel for ZEN to pass through. However, the bolt in previously studied structures is tighter than that in NbZHD. Despite this, convincing evidence has yet to demonstrate that ZEN enters the catalytic core through the bolt, although it is possible that the space around the bolt could play a role in this process. Clear changes in catalytic efficiency have been observed in mutant D157K and E171K of ZHD101, whose corresponding bolt is V153-H173 (colored olive), where D157K decreased and E171K increased in terms of k_cat_/K_M_ through their respective charge changes [25]. Theoretical research has shown that mutant V153H can increase the interaction between M241 and Y245, inducing the sidechain of H242, one part of the catalytic triad [24], and positioning it into a positive catalytic conformation [26] (Appendix A). The bolt of ZHD607, characterized by V156-A176 (colored red), displays similar features to those of ZHD101 (Appendix A). Mutations I160Y and G163N in ZHD607 have a notable impact on enzymatic activity, with G163N decreasing activity while I160Y increases it 3.4-fold. According to Yao et al., the I160Y mutation alters the positions of I160 and P135, reducing the distance between Y160 and P135 and stabilizing ZEN in the catalytic pocket, as observed through molecular dynamics simulation [13].

The calculation of free energy in molecular dynamics can be divided into two types: two-end-state calculation and enhanced sampling calculation. Molecular mechanics/Poisson–Boltzmann (Generalized-Born) surface area (MMPBSA) is a method of two-end-state calculation, while umbrella sampling is a method of enhanced sampling calculation. MMPBSA is one of the methods of two-end-state calculation, while umbrella sampling is one of the methods of enhanced sampling calculation. The mean binding energy of NbZHD is −8.45 kcal/mol. The binding energy of D45A and R139A mutants are −8.13 kcal/mol and −8.76 kcal/mol, respectively. However, the PMF calculation yielded a different result from MMPBSA, which was −1.95 kcal/mol. In the QM/MM study of RmZHD, the binding free energy from the docking state to the end of the catalytic reaction is −6 kcal/mol, which is more consistent with our result from MMPBSA.

## 5. Conclusions

This study aimed to develop a pipeline for detecting the motion paradigm of the zearalenone hydrolase protein family. Eight hub genes were selected through identity analysis, and molecular dynamics simulations conducted on the residues 139–222 from ZHD indicated the allosteric pathway was Q183 → E177 → P174 → L154 → L149 → Q146 → I202. Umbrella sampling was used to capture the instant docking dynamic simulation for NbZHD, enabling the identification of the allosteric pathway at this time point: M207 → R139 → V145 → Q152 at the cap domain and D45 → A38 → S62 → L66 at the hydrolase domain. MMPBSA was used to validate the binding energy of some residues on the pathway, including alanine mutants. The binding energy of D45A was −8.13 kcal/mol, R139A was −8.76 kcal/mol, and the unaltered binding energy was −8.45 kcal/mol. Our study revealed two distinct stages in the behavior of ZHD, before and after ZEN docking. Specifically, before the docking phase, the cap domain may act as a hemostatic tape upon the skin-base of the hydrolase domain similarly to the open-to-close or the close-to-open process of the cavity. During the docking phase, the substrate zearalenone (ZEN) passes between the “bread” of the cap and hydrolase domains through a tunnel opposite the cavity. The zearalenone hydrolase (ZHD) then facilitates the rotation of the cap and hydrolase domains in opposing directions, resembling a cube. The findings of this study can improve our understanding of how ZHD moves in the liquid phase.

## Figures and Tables

**Figure 1 molecules-28-04545-f001:**
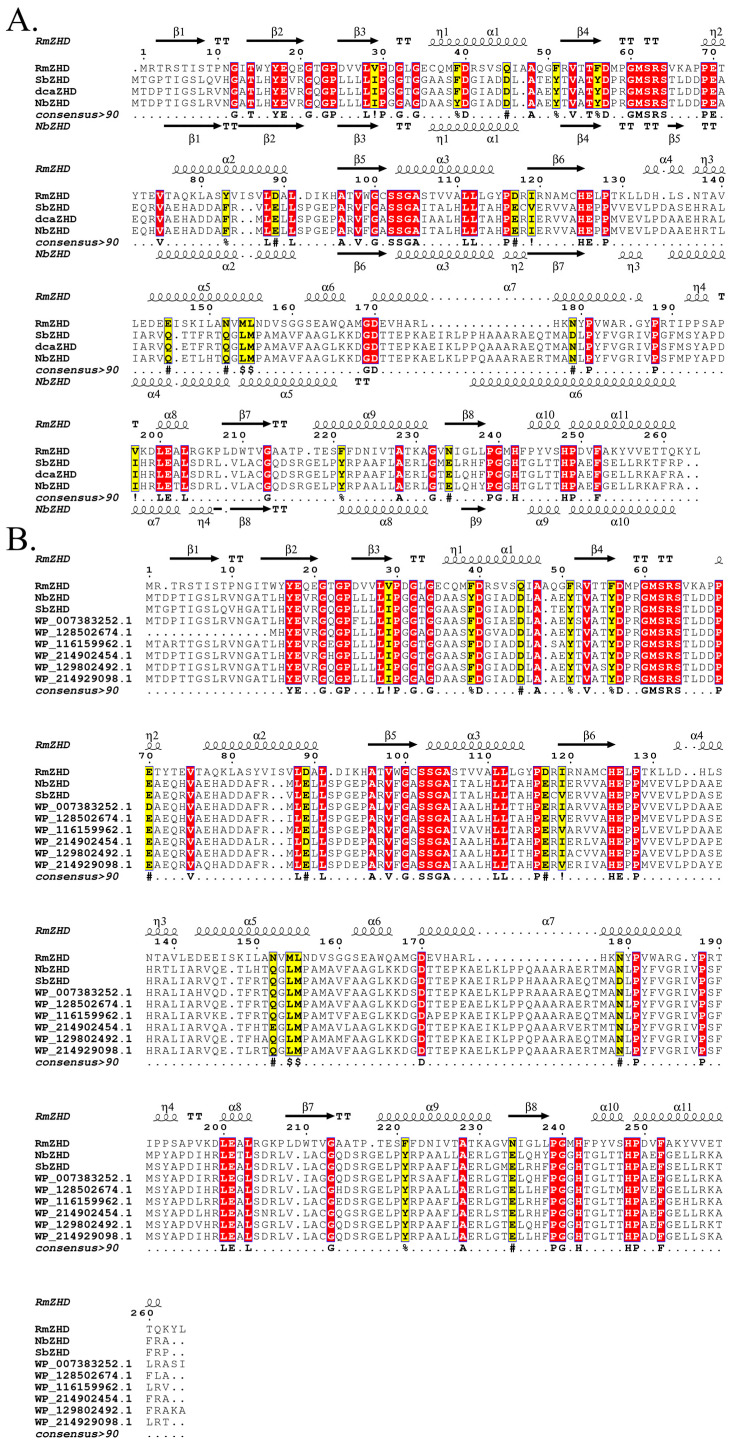
Multiple sequence alignment of representing zearalenone hydrolase and hub enzymes. (**A**) Multiple Sequence Alignment (MSA) of RmZHD, NbZHD, SbZHD and a consensus sequence dcaZHD. (**B**) MSA of RmZHD and 8 hub genes.

**Figure 2 molecules-28-04545-f002:**
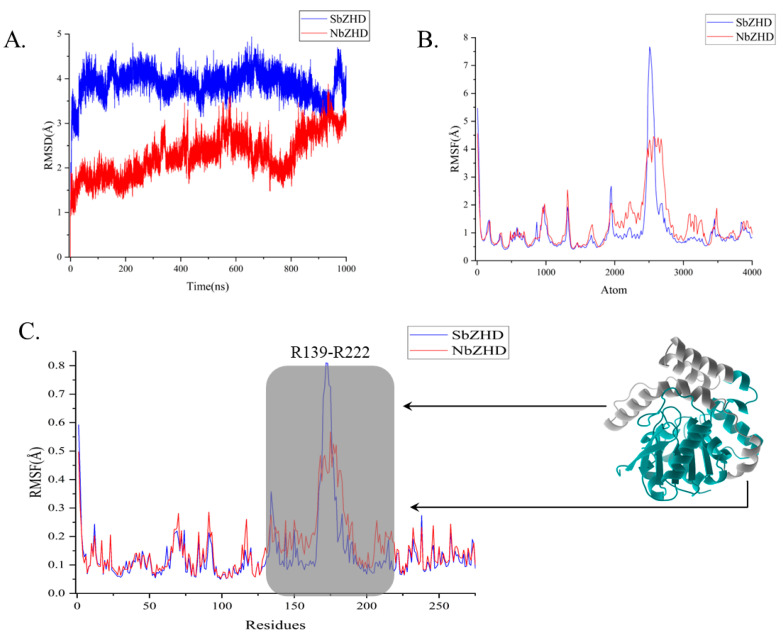
RMSD and RMSF analysis of 1 μs simulation calculated for motion-residue sampling. (**A**) RMSD of 1 μs simulation of NbZHD and SbZHD. (**B**) RMSF of 1 μs simulation of NbZHD and SbZHD at atomic level. (**C**) RMSF of 1 μs simulation of NbZHD and SbZHD at residues level and the grey perspective marked zone; i.e., R139–R222 is selected for motion-residue sampling.

**Figure 3 molecules-28-04545-f003:**
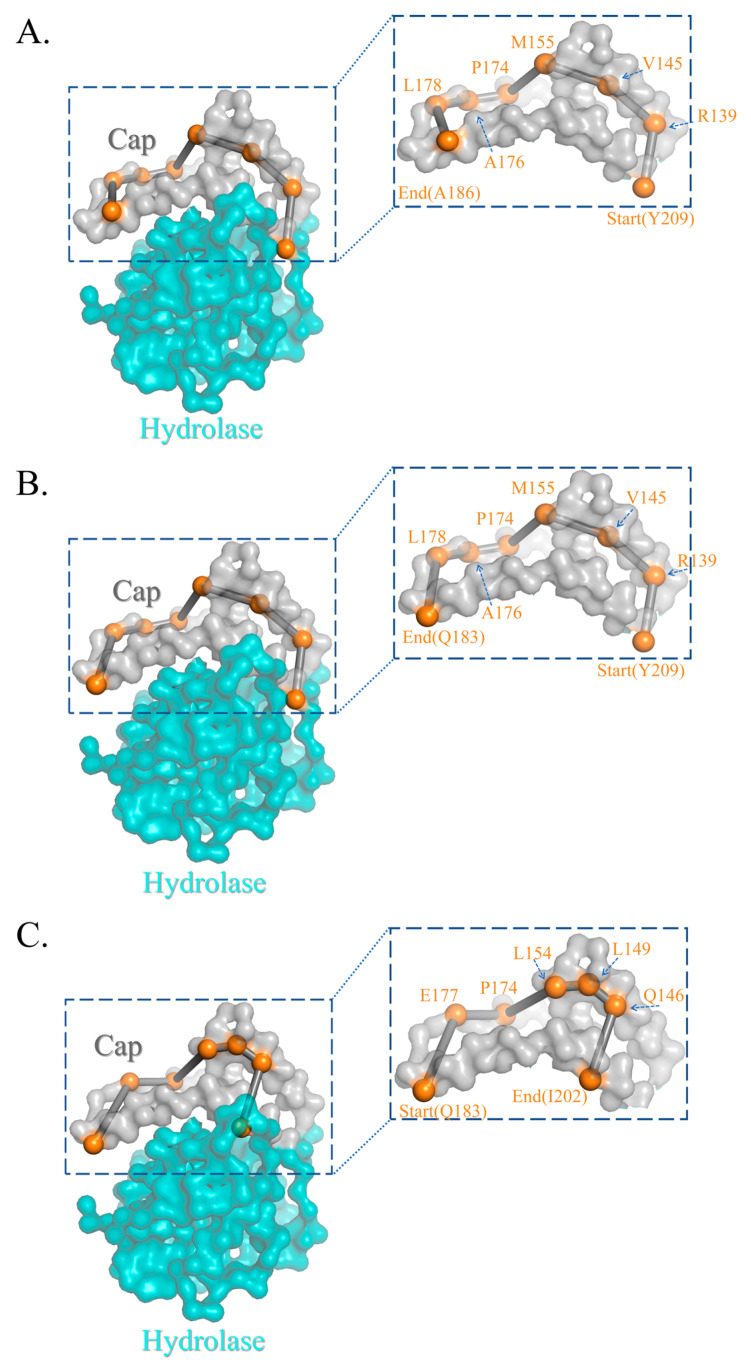
The diagrams indicate the allosteric pathway and the corresponding amino acids from 1 μs molecular dynamics simulation. The domain in teal is the hydrolase domain locating the sites of 1–138 and 223–278 while the domain in grey is the cap domain locating the sites of 139–222. The orange spheres are refered as the pathway of motion residues. (**A**,**B**) Two pathways that contain 8 residues. (**C**) A pathway that contains 7 residues with the highest probability.

**Figure 4 molecules-28-04545-f004:**
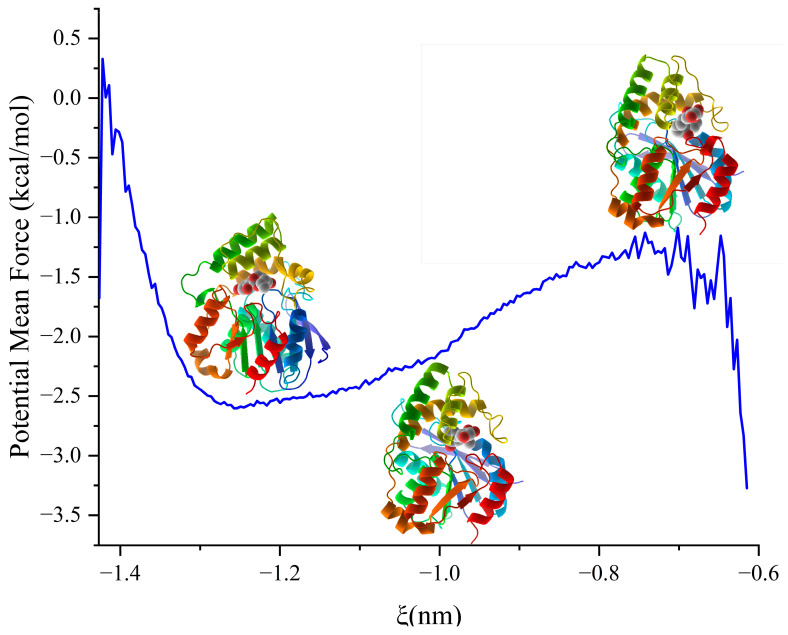
PMF analysis of pulling process. The left one is when the zearalenone adapting the catalytic center, the middle one is when the complex state in the inflection point, and the right one is when the zearalenone in the releasing state.

**Figure 5 molecules-28-04545-f005:**
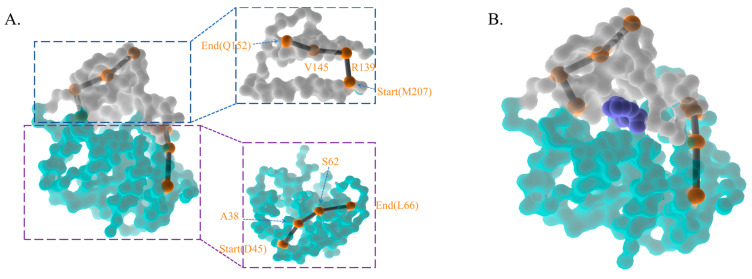
The diagrams indicate the allosteric pathway and the corresponding amino acids from umbrella sampling. The domain in teal is the hydrolase domain locating the sites of 1–138 and 223–278 while the domain in grey is the cap domain locating the sites of 139–222. The orange spheres are refered as the pathway of motion residues. The deepblue module refers to the ZEN. (**A**) The highest probability pathway from concatenated windows. (**B**) The frontside with the ligand.

**Figure 6 molecules-28-04545-f006:**
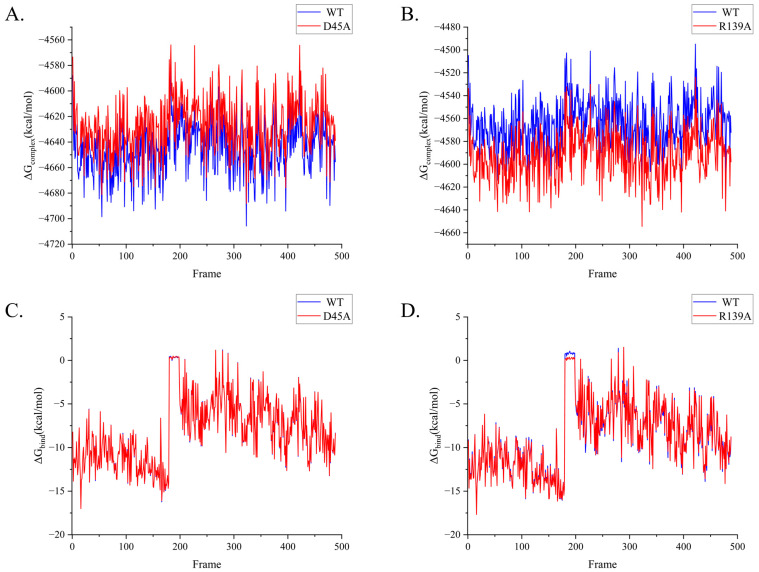
The energy variation of the wild type and mutant enzymes. (**A**,**B**) ΔG of complex from two mutants, D45A and R139A. (**C**,**D**) ΔG of binding from two mutants, D45A and R139A.

**Figure 7 molecules-28-04545-f007:**
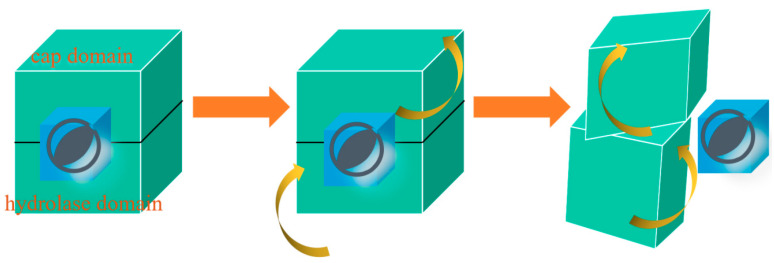
The generative vision of the paradigm. We consider this protein as consisting of two modules, both colored teal. The upper module is referred to as the cap domain, while the lower module is referred to as the hydrolase domain. The blue cube positioned in the middle represents ZEN. The relative motion between the upper and lower modules is facilitated by distinct directional arrows associated with each module, indicating their respective movements. At first, the ligand is docking in the ZHD, and the cap domain is counterclockwise rotating while the hydrolase domain is clockwise. After the emission of the ligand, both domains of ZHD will rotate in the opposite direction.

**Figure 8 molecules-28-04545-f008:**
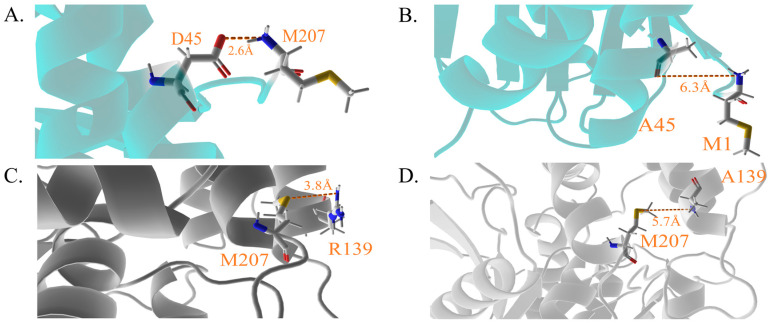
The diagrams indicate the mutant sites and corresponding residues. (**A**,**B**) The residues interface of D45 and its mutant A45 with M1. (**C**,**D**) The residues interface of R139 and its mutant A139 with M207. The shorter the ionic bonds, the stronger the binding.

**Figure 9 molecules-28-04545-f009:**
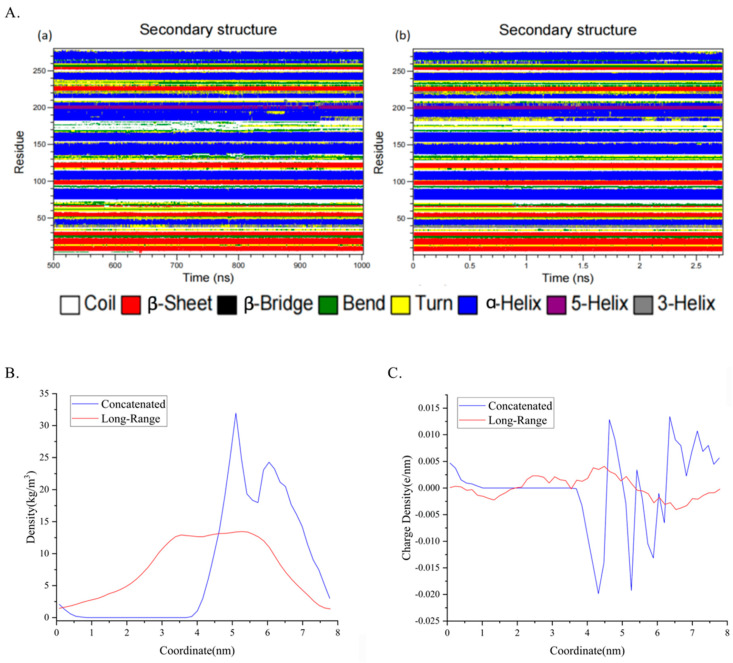
Secondary structure analysis inspects the local shifting including density and charge density. (**A**) Secondary structure analysis across the time scale, (**a**) belongs to the late 500 ns simulation of long-range simulation and (**b**) belongs to the concatenated window with 1361 frames, each frame separated by 2 ps, summing 2.7 ns. (**B**,**C**) Mass density of residues 170 to residues 190 along the coordinate and charge density of them based on the comparison of the concatenated one and the long-range one.

**Table 1 molecules-28-04545-t001:** The table shows 11 pathways that contains 7 residues with over 99.9% probability and 2 pathways with 8 residues.

Item	Number of AA	Pathway	Probability
1	8	209 → 139 → 145 → 155 → 174 → 176 → 178 → 183	85.778%
2	8	209 → 139 → 145 → 155 → 174 → 176 → 178 → 186	81.800%
3	7	183 → 177 → 174 → 154 → 149 → 146 → 202	99.999%
4	7	184 → 177 → 174 → 154 → 149 → 146 → 202	99.974%
5	7	181 → 177 → 174 → 154 → 149 → 146 → 202	99.971%
6	7	184 → 176 → 175 → 154 → 150 → 146 → 202	99.935%
7	7	182 → 176 → 175 → 154 → 150 → 146 → 202	99.935%
8	7	184 → 176 → 174 → 154 → 149 → 146 → 202	99.922%
9	7	182 → 176 → 174 → 154 → 149 → 146 → 202	99.922%
10	7	184 → 176 → 175 → 154 → 149 → 146 → 202	99.914%
11	7	182 → 176 → 175 → 154 → 149 → 146 → 202	99.913%
12	7	181 → 176 → 175 → 154 → 150 → 146 → 202	99.907%
13	7	186 → 176 → 175 → 154 → 149 → 146 → 202	99.901%

**Table 2 molecules-28-04545-t002:** Energy analysis of NbZHD and its mutants, D45A and R139A. NbZHD(1) stands for the first-time calculation with D45A; NbZHD(2) stands for the second-time calculation with R139A; NbZHD is the average of NbZHD(1) and NbZHD(2). The below data are reported in the unit of kcal/mol.

Enzyme	ΔVDW	ΔEEL	ΔE_MM_	ΔG_SOLV_	ΔG_GAS_	ΔG_bind_
NbZHD(1)	−31.84 ± 7.06	−2.97 ± 1.03	−34.81 ± 8.09	26.68 ± 5.42	−34.82 ± 7.74	−8.14 ± 3.74
D45A	−31.84 ± 7.06	−3.00 ± 1.04	−34.84 ± 8.10	26.71 ± 5.43	−34.84 ± 7.75	−8.13 ± 3.74
NbZHD(2)	−31.84 ± 7.06	−2.97 ± 1.03	−34.81 ± 8.09	26.07 ± 5.21	−34.82 ± 7.74	−8.75 ± 3.97
R139A	−31.84 ± 7.06	−2.96 ± 1.03	−34.80 ± 8.09	26.04 ± 5.32	−34.80 ± 7.74	−8.76 ± 3.94
NbZHD	−31.84 ± 7.06	−2.97 ± 1.03	−34.81 ± 8.09	26.38 ± 5.32	−34.82 ± 7.74	−8.45 ± 3.86

## Data Availability

The data presented in this study are available on request from the corresponding author. The data are not publicly available due to interests.

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
