# Peer review of "The Motion Paradigm of Pre-Dock Zearalenone Hydrolase Predictions with Molecular Dynamics and the Docking Phase with Umbrella Sampling"

_molecules, 2023, doi:10.3390/molecules28114545_

Round 1
Reviewer 1 Report
The study presented aims to develop a pipeline for detecting the motion paradigm of the zearalenone hydrolase (ZHD) protein family
The findings mentioned regarding the behavior of ZHD before and after ZEN docking are intriguing. However, more elaboration is needed to understand the specific characteristics and implications of ZHD acting like a "hemostatic tape" before docking and a "square sandwich" after docking. How do these behaviors relate to the protein's function or structure?
The significance and potential implications of the identified allosteric pathways should be discussed. How do these pathways relate to the overall function or regulation of ZHD? Are they conserved among related proteins?
The passage contains technical terms and acronyms (e.g., ZHD, ZEN, MMPBSA, NbZHD) that should be defined or explained upon their first mention to ensure clarity for readers who may not be familiar with the specific terminology.
Thorough proof reading has to be performed with native english person
Author Response
Dear Reviewer:
The following issues you mentioned have been addressed in the revised manuscript.
Question1: The findings mentioned regarding the behavior of ZHD before and after ZEN docking are intriguing. However, more elaboration is needed to understand the specific characteristics and implications of ZHD acting like a "hemostatic tape" before docking and a "square sandwich" after docking. How do these behaviors relate to the protein's function or structure?
Response1: In response to the first question, the crude paraphrasing refers to a motion pathway of structural movement during protein catalysis, which probably also involves the function of the protein. Specifically, before the docking phase, the cap domain may act as a hemostatic tape upon the skin-base the hydrolase domain similar to the open to close or the close to open process of the cavity. During the docking phase, the substrate zearalenone (ZEN) passes between the “bread” of the cap and hydrolase domains through a tunnel opposite the cavity. The zearalenone hydrolase (ZHD) then facilitates the rotation of the cap and hydrolase domains in opposing directions, resembling a cube.
Question2: The significance and potential implications of the identified allosteric pathways should be discussed. How do these pathways relate to the overall function or regulation of ZHD? Are they conserved among related proteins?
Response2:
In regards to the second question, the predicted residue pathways are comparatively evolutionary conserved as they merely provide one possible perspective on the dynamic paradigm of the ZHD; minor local changes could result in different residue pathways. The conservative information is provided in Table S1. Specifically, the pathways derived from the docking phase represent a potential logic for the movement during the actual catalytic process, whereas those from before the docking phase suggest a rationale for how the ZHD recruits the ZEN.
Question3: The passage contains technical terms and acronyms (e.g., ZHD, ZEN, MMPBSA, NbZHD) that should be defined or explained upon their first mention to ensure clarity for readers who may not be familiar with the specific terminology.
Response3: In response to the thirdquestion, I have addressed the issue in the new manuscript.
I believe these modifications and improvements have enhanced the quality and comprehensibility of the manuscript. I sincerely hope that you will acknowledge these changes and allow me to resubmit the revised version to your journal.
Thank you once again for your patience and cooperation. I look forward to your evaluation of the revised manuscript and the opportunity to publish this research in your esteemed journal.
Thank you for your attention.
Best regards,
Xizhi Hong
Reviewer 2 Report
Review Report:
In the manuscript, Hong et al. demonstrate a pipeline for detecting the motion paradigm of zearalenone hydrolase (ZHD) protein family that has a substantial capability to degrade one of the most prevalent mycotoxins, zearalenone (ZHN). During the MD simulation, they were able to identify the allosteric pathway of the protein involving a neural relational inference model. This leads to the analysis of several residues, which revealed two distinct stages of ZHD before and after ZEN docking, hemostatic tape like followed by square sandwich afterwards.
I find the work important in the context of ZHN degradation mechanism. The manuscript is thorough, well-written and presented. I will recommend the publication of the present work essentially as is after addressing some minor modifications. I have used the following abbreviations, P-page number and L-line number to point out my comments.
Specific Comments:
1. P3-L98: I encourage the authors to introduce the abbreviation for the hydrogen bonds as h-bonds.
2. P3-L99: The unit should be checked.
3. P3-L116: I recommend the authors to rewrite this sentence for better readability.
Author Response
Dear Reviewer:
The following issues you mentioned have been addressed in the revised manuscript.
Question1: P3-L98: I encourage the authors to introduce the abbreviation for the hydrogen bonds as h-bonds.
Response1: The identified issues have been effectively addressed and resolved in accordance with the revisions made in the updated manuscript.
Question2: P3-L99: The unit should be checked.
Response2: The incorrected units have been checked and addressed and resolved in accordance with the revisions made in the updated manuscript.
Question3: P3-L116: I recommend the authors to rewrite this sentence for better readability.
Response3: The readability of the sentence you mentioned has been upgrated and resolved in accordance with the revisions made in the updated manuscript.
I believe these modifications and improvements have enhanced the quality and comprehensibility of the manuscript. I sincerely hope that you will acknowledge these changes and allow me to resubmit the revised version to your journal.
Thank you once again for your patience and cooperation. I look forward to your evaluation of the revised manuscript and the opportunity to publish this research in your esteemed journal.
Thank you for your attention.
Best regards,
Xizhi Hong